# Sub-Daily Simulation of Mountain Flood Processes Based on the Modified Soil Water Assessment Tool (SWAT) Model

**DOI:** 10.3390/ijerph16173118

**Published:** 2019-08-27

**Authors:** Yongchao Duan, Fanhao Meng, Tie Liu, Yue Huang, Min Luo, Wei Xing, Philippe De Maeyer

**Affiliations:** 1State Key Laboratory of Desert and Oasis Ecology, Xinjiang Institute of Ecology and Geography, Chinese Academy of Sciences, Urumqi 830011, China; 2Key Laboratory of GIS & RS Application Xinjiang Uygur Autonomous Region, Urumqi 830011, China; 3University of Chinese Academy of Sciences, Beijing 100049, China; 4Department of Geography, Ghent University, Ghent 9000, Belgium; 5Sino-Belgian Joint Laboratory of Geo-Information, Urumqi 830011, China; 6Sino-Belgian Joint Laboratory of Geo-Information, Ghent 9000, Belgium; 7Inner Mongolia Normal University, Hohhot 010022, China

**Keywords:** sub-daily, flood processes, accumulated temperature (AT), degree-day factor, SWAT

## Abstract

Floods not only provide a large amount of water resources, but they also cause serious disasters. Although there have been numerous hydrological studies on flood processes, most of these investigations were based on rainfall-type floods in plain areas. Few studies have examined high temporal resolution snowmelt floods in high-altitude mountainous areas. The Soil Water Assessment Tool (SWAT) model is a typical semi-distributed, hydrological model widely used in runoff and water quality simulations. The degree-day factor method used in SWAT utilizes only the average daily temperature as the criterion of snow melting and ignores the influence of accumulated temperature. Therefore, the influence of accumulated temperature on snowmelt was added by increasing the discriminating conditions of rain and snow, making that more suitable for the simulation of snowmelt processes in high-altitude mountainous areas. On the basis of the daily scale, the simulation of the flood process was modeled on an hourly scale. This research compared the results before and after the modification and revealed that the peak error decreased by 77% and the time error was reduced from ±11 h to ±1 h. This study provides an important reference for flood simulation and forecasting in mountainous areas.

## 1. Introduction

Flooding is one of the most common and feared natural disasters around the world. The losses caused by floods account for approximately 40% of the losses of all natural disasters [1]. The destructive power of floods causes great losses to peoples’ lives and their property security. Based on the source of the flood, flooding can be classified as either rainfall, snowmelt, or mixed floods [2,3,4,5,6,7]. In flat plain areas, since there is no water supply from melting snow and melting ice, the water originates primarily from rainfall and the flooding in these areas is mainly rainfall flooding [2,5]. In high-altitude, mountainous areas, however, perennial snow cover and glacier cover often lead to snowmelt and mixed floods [8]. Today, under the influence of climate change and an increasing population, the losses and hazards caused by mountain torrents are becoming increasingly prominent and serious. Given this situation, it is an urgent responsibility of government departments to ensure the safety of residents and property when floods occur [1,9]. Therefore, there is a crucial need for flood disaster simulation and prevention models to be developed and improved. The premise of the accurate prediction of flood events, however, is to study different types of flood processes [10]. Hence, the simulation of flood processes under high temporal resolution is becoming extremely important [11].

With this emphasis on flood process simulation, various models have been modified and applied to flood simulation [12,13,14,15,16,17], including empirical models, physically-based models, conceptual models, semi-distributed and distributed hydrological models. The computational formulas of these models cover complex energy conservation, degree-day factor methods, and water balance methods [18,19,20,21]. Although these models are accepted for various reasons, including their simple structure [22], high computational efficiency, and smaller data requirements, they all insufficiently account for the spatial heterogeneity of catchment areas and the impact of complex terrain. Therefore, more and more distributed and semi-distributed hydrological models have been applied to flood process simulation [23,24,25]. The TOPMODEL was first used for flood simulation. The variable infiltration capacity (VIC) model was also used to flood process simulations [26,27]. The Xin’anjiang (XAJ) model was the first hydrological model developed in China and used for flood simulation, which achieved a high accuracy in the Huaihe River Basin. Many other models have achieved varying results in flood simulation, such as the storm water management model (SWMM), hydrological, MIKE Systeme Hydrologique Europeen (MIKE SHE) [28], and the MARINE event-based model [29,30,31]. To make the simulation of evaporation more accurate under conditions of low relative humidity, low temperature, and strong winds, a mass and energy balance snowmelt point model was developed by Herrero et al. using a new atmospheric emissivity expression [32]. The fast all-season soil strength (FASST) model was applied to calculate the amount of snowmelt in the alpine regions of China by Yu et al. [33]. They found that many parameters had to be evaluated when appraising and verifying this method, which led to a huge workload. This problem was finally solved by combining the FASST model with the soil water assessment tool (SWAT) model [34,35,36].

The SWAT model [37,38,39] is a typical physically-based distributed hydrological model [40,41]. To calculate total surface runoff and snowmelt, the Soil Conservation Service (SCS) runoff curve number method and the degree-day factor method were used [35,42,43]. The SWAT model can be integrated perfectly with a geographic information system (GIS), which demonstrates its great advantages and applicability in flood process simulation. In addition, the model offers significant advantages in runoff simulation, water quality simulation, and sediment migration. When the original SWAT model was applied to the runoff simulation in high-altitude mountainous areas, however, some shortcomings emerged. As a condition for the occurrence of snowmelt, the original degree-day factor method simply compared the threshold value of snowmelt temperature with the average daily temperature, ignoring the influence of accumulated temperature. Because the accumulation of temperature has a more pronounced effect on snowmelt in high-altitude mountainous areas, it is more reasonable to use the accumulated temperature threshold as the snowmelt condition. At the same time, when there is discrimination among precipitation types, the original model is also inadequate. Some researchers improved the model after recognizing these shortcomings. Meng et al. built an algorithm that used elevation to yield distribution temperature and precipitation changes with elevation [44]. Luo et al. added an ice-melting module to the SWAT model [45]. These improvements, however, were also based on the traditional method.

When simulating flood processes, the accuracy of the model on the daily scale is important. For snowmelt flood types in mountainous areas with snow cover, the daily scale cannot meet the accuracy requirements of flood process simulation. Therefore, it is necessary to simulate the flood processes on a finer time scale. The SWAT model can provide a runoff simulation on a sub-daily scale [38,46,47], a daily timescale for continuous simulations and a sub-daily timescale for event-based flood simulation. The dimensionless unit hydrograph (UH) method was employed in the SWAT model exhibits a triangular shape [48,49,50]. When the precipitation data are input into the model, the daily precipitation is mainly used to simulate and represent the surface runoff, and it is continuous. However, the input of precipitation data on the hourly or even minute scales is mainly used to simulate a flood process, which does not require continuous simulation and shortens the calculation time of the model. Dan et al. improved the event-based SWAT model by using a sub-daily scale, thereby greatly improving the simulation results [47]. Zheng et al. improved the model algorithm to run the SWAT model on a sub-daily, or even sub-hourly, time scale, which created a continuous simulation with higher time precision. At present, sub-daily simulations have primarily been applied to the study of rainfall flooding. These simulations are seldom used in snowmelt floods in high-altitude mountainous areas; therefore, this study provides a reference for related research in this area. On the basis of the existing flood process research, and in light of the deficiencies in snowmelt flood simulations in mountainous areas, the original SWAT model was modified and simulations were conducted on daily and hourly scales. By increasing the ability to determine the type of precipitation, the precipitation classification involved in this model calculation was more accurate. When calculating snowmelt, the effect of accumulated temperature on snow melting was fully considered, which modified the snow melting threshold. Using the daily scale as the basis of our simulation, the sub-daily scale was simulated and compared to the expression of the modified model. This modified the performance of the model for the snowmelt flood process in high-altitude mountainous areas. Based on the existing research, the SWAT model is modified to make it more suitable for high altitude mountain areas. The simulation of flood process under high time resolution is realized, which can provide a new understanding of the mechanism of flood occurrence and the induced factors, establish a research basis for future flood forecasting, and provide technical support for flood prevention.

## 2. Study Area and Data

### 2.1. Study Area

One of the areas in China with a high incidence of flooding, and where significant flood events occur every year, is Xinjiang. China is a vast country with complex topography and large climate differences [22,51]. Western China contains numerous high mountains, and many of the sources of large rivers are located in these mountains. In these mid- to high-latitude regions, snowmelt provides 80% of the runoff [52] and rich water resources at the same time [53,54,55]. As such, it relieves pressure on all kinds of water needs to a certain extent [45,56]. Although snowmelt provides water resources, it also causes flood disasters. According to previous research, mountain torrents account for 70% of the casualties and more than 50% of the economic losses caused by floods [57,58,59,60]. The region’s many mountainous areas and the complex terrain in Xinjiang [61,62,63] also contribute to the distinctive climatic conditions and hydrological events. In high-altitude mountainous areas with scarce vegetation, snow falls more in the winter and precipitation is reduced in the spring [64]. With the general temperature increase in the spring, surface temperatures increase rapidly, snow begins to melt, and river runoff increases, resulting in snowmelt floods [65,66]. As night falls, however, the temperature drops, and the process of snow melting decreases or even stops [67]. This situation has resulted in the unique snow-melting floods, characteristic of the mountainous areas of Xinjiang, which have been identified as a “one peak, one day” phenomenon [68,69]. This study took place at the Tizinafu River Basin (TRB) in Xinjiang, China which was the research area. It provides a scientific reference for subsequent research on the further improvement of flood simulation accuracy.

The Tizinafu River Basin (TRB) (Figure 1) is situated in the western part of China and the southwestern part of Xinjiang. The Keerake Daban and Yanggai Daban were located on the northern slopes of the Kunlun Mountains and also the source of the Tizinafu River. The total length of the basin is about 335 km and the area of the basin is approximately 5600 km^2^. The TRB is located in the Kunlun Mountains, and features limited vegetation, numerous mountainous areas, and large fluctuations in altitude, ranging from 1476–6320 m. The lower-elevation plain area has reduced precipitation, while the mountainous area is affected by a microclimate, and precipitation is a little more than that in the plain area. The average daily precipitation and temperature in the basin from 2011–2014 were 6.71 °C and 0.6 mm, respectively. The average temperature in the basin is 3.28 °C in spring, 15.1 °C in summer, 12.34 °C in autumn and −2.47 °C in winter.

The TRB features a large area of snow and glaciers in the high-elevation regions, which explains why snowmelt floods often occur in this basin. From May–September, the TRB normally experiences a high incidence of floods. Due to the limited vegetation and numerous rocks, as temperatures rise each May, the surface temperature increases rapidly. As a result, the snowmelt and runoff supply to the river increase rapidly as well, frequently leading to flooding [70]. In addition, summer precipitation commonly triggers floods. From October–February (known as the dry season), the temperature drops, snowfall increases in mountainous areas, snow melting stops, and the rivers dry up. During the research period, the average snow-cover area during the snow-melting season was 103.47 km^2^ and that area of the non-snowmelt season was 717.81 km^2^. According to the analysis and statistics of the observed hydrological data, the annual average runoff of the TRB was about 35.22 m^3^/s.

### 2.2. Materials

The SWAT model was constructed to utilize many basic input data to simulate the hydrological process. The model was calibrated and verified using measured data, including (Digital Elevation Model) DEM base data, soil-type data, surface cover-type data, and meteorological data. In this study, the basic DEM data with a spatial resolution of 30 m was adopted from the(Shuttle Radar Topography Mission) SRTM official website (http://srtm.csi.cgiar.org/). The soil type data also had a 30-m resolution and was downloaded from the China Soil Category Data Network (https://geodata.pku.edu.cn/). The 30-m resolution land use and land cover (LUCC) data were obtained from the visual interpretation of imagery (https://www.usgs.gov/products/data-and-tools/real-time-data/remote-land-sensing-and-landsat). The precipitation event data and flood event statistics were obtained from the statistical yearbooks of relevant local departments. The MODA10A2 (2013–2014) is the important snow product and the model input data, which were obtained from the corresponding temperature threshold.

The meteorological and hydrological data obtained and used in this study were from the Momoke, Xihexiu, and Kudi meteorological stations, and the Jiangka hydrological station. As shown in Figure 1, the three meteorological stations were located at elevations of 1863 m, 2826 m, and 3067 m, respectively. The observational data from the meteorological stations used in this study began with the establishment of the stations in 2010, continuing until 2019. The data included both daily-scale data and minute-level data, including maximum temperature data, minimum temperature data, wind speed and direction data, air humidity data, soil humidity data, surface radiation data, and other important meteorological data that were significant for model building and calibration. According to the calculation and statistics of the temperature data observed at the three meteorological stations, the daily maximum, minimum, and average temperatures of the basin during the research period were 13.36 °C, 0.89 °C, and 7.13 °C, respectively. 

## 3. Methods

On the basis of the original model, the effect of the accumulated temperature on snow melting was fully considered. The daily accumulated temperature was calculated according to the temperature integral method and the influence of this factor was then added to the model calculation. The runoff result was simulated and compared on a daily scale, and the model simulation’s results were evaluated according to the Nash–Sutcliffe efficiency (NSE), determination coefficient (R^2^), and percent bias (PBIAS). Based on the daily scale, the simulation results for flood processes under high temporal resolution were obtained from the hourly scale simulation.

### 3.1. Modification of Sub-Daily Flood Process Simulation 

In this study, the modification of the original SWAT model was primarily focused on two points. First, was the new added determination of precipitation type, and second was the modified degree-day factor based on the traditional one. Both the determination of precipitation type and the calculation of snowmelt amount cannot be separated from the influence of energy. Actually, temperature was the most important factor affecting energy accumulation and dissipation. Therefore, it was reasonable to modify the snowmelt module by calculating the accumulated temperature and adding it to the model.

This study’s research ideas are illustrated in Figure 2, including the establishment of the daily model, modification of the snowmelt module (red box), and simulation of hourly flood events. First, the daily scale model was established, and the snowmelt module was modified. Then, after calibration and verification of the daily scale model, the daily scale was reduced to the hourly scale for additional calibration and verification. The precipitation types were determined and separated by increasing the cumulative temperature determination conditions (Section 3.2) and calculating each type separately. When the precipitation type was snowfall, it was added to the snow cover for the snow-melting calculation. When determining the conditions of snowmelt occurrence, the calculation of snowmelt can be carried out only when the temperature exceeds the set threshold by increasing the maximum temperature and accumulated temperature. In this way, the discharge simulation based on the daily scale was completed. On this basis, the sub-daily scale simulation was carried out in order to realize the simulation study of flood processes on the hourly scale driven by hourly precipitation data, by increasing the time scale of rainfall data, the calculation step of the model is increased to the hour scale, and the calculation of snowmelt is increased from the day scale to the hour scale, thereby improving the accuracy of the flood process simulation under a high time resolution.

### 3.2. Calculation of Accumulated Temperature

In the original model, a simple comparison of the average temperature with the temperature threshold was applied for the determination of precipitation type [66]. This approach had a certain rationality in plain areas, but when applied to high-altitude mountainous areas, its accuracy was questionable. In addition, the mountainous areas were generally affected by various microclimates; therefore, the method of determining precipitation type simply on the basis of daily temperature required improvement.

Usually, the ambient temperature rises in the morning, reaches a maximum at midday, and then slowly declines. The general trend of change can be approximated as a sinusoidal curve. Figure 3 compares this diurnal variation of measured temperature with the simulated approximate sinusoidal curve.

In terms of maximum and minimum temperature variations, there are three cases of temperature variations throughout the year, as shown in Figure 4. Curve T1 is the case in which the maximum and minimum daily temperature is greater than 0 °C; in the case of T2, the maximum temperature is >0 °C, but the minimum temperature is <0 °C; the curve labelled T3 illustrates the case in which the maximum temperature is <0 °C. Actually, when the influence of accumulated temperature on precipitation morphology and snowmelt is considered, the calculated accumulated temperature will only be meaningful when the temperature is >0 °C. Therefore, when the accumulated temperature was calculated, only the T1 and T2 curves were considered, and the 24 h of the day were represented by the 0–π radians.

The temperature at different times of day can be calculated by the following formula:(1)$$T_{day}=\left(T_{mx}-T_{mn}\right)\sin t+T_{mn}\;\;\;\;\;\;\;0\leq t\leq \pi$$
the formula for calculating the accumulated temperature was as follows:(2)$$T\;=\;\{\begin{array}{c} \int_{0}^{\pi }\left(T_{mx}-T_{mn}\right)\sin t+T_{mn\;}dt,\;t\in \left(0,\pi \right)\\ \int_{\sin^{-1}(\frac{-T_{mn\;}}{T_{mx}-T_{mn}})}^{\pi -\sin^{-1}(\frac{-T_{mn\;}}{T_{mx}-T_{mn}})}T_{mx}\sin tdt,\;t\in \left({sin}^{-1}(\frac{-T_{mn\;}}{T_{mx}-T_{mn}}\right),\pi -{sin}^{-1}(\frac{-T_{mn\;}}{T_{mx}-T_{mn}})) \end{array},$$

When the temperature change was curve T1, the formula for calculating the accumulated temperature was the first formula of the Formula (2); when the temperature change was curve T2, the accumulated temperature was the second one of the Formula (2).

Where $T_{day}$ is the temperature at different times, T is the daily accumulated temperature, $T_{mx}$ is the daily maximum temperature, $T_{\mathrm{mn}}$ is the daily minimum temperature, *t* is the radian corresponding to the moment of the day, and ${sin}^{-1}(\frac{-T_{mn\;}}{T_{mx}-T_{mn}})$, $\pi -{sin}^{-1}(\frac{-T_{mn\;}}{T_{mx}-T_{mn}})$ are the radians corresponding to 0 ℃, which were gotten by making Formula (1) zero.

### 3.3. Calibration, Validation and Sensitivity

The sequential uncertainty fitting (SUFI-2) [71,72,73,74] algorithm of the Soil and Water Assessment Tool Calibration and Uncertainty Procedure (SWAT-CUP) [73,75] was applied for the parameter calibration, model validation, and parameter sensitivity analysis [76,77,78,79,80]. During the simulation of the model, 2011–2012 was taken as the warm-up time, 2013 as the calibration period, and 2014 as the verification period. The NSE, R^2^, and PBIAS were applied to evaluate the simulation results.

The NSE was mainly utilized to evaluate the fitting level between the modelling simulation values and the observations. The range was from −∞ to 1. The closer to 1, the better. For the convenience of evaluation, the model’s performance was divided into four grades: “unsatisfactory”(NSE ≤ 0.5), “satisfactory” (0.5 < NSE ≤ 0.65), “good” (0.65 < NSE ≤ 0.75) or “very good” (0.75 < NSE ≤ 1.0), respectively [81]. The method for calculating the NSE value was as follows:(3)$$NSE=1-\frac{\sum_{i=1}^{n}{\left(Q_{obs,i}-Q_{sim,i}\right)}^{2}}{\sum_{i=1}^{n}{\left(Q_{obs,i}-{\bar{Q}}_{obs,i}\right)}^{2}}$$
(4)$$R^{2}={\left[\frac{\sum_{i=1}^{n}(Q_{sim,i}-{\bar{Q}}_{sim,i})\left(Q_{obs,i}-{\bar{Q}}_{obs,i}\right)}{\sqrt{\sum_{i=1}^{n}{\left(Q_{sim,i}-{\bar{Q}}_{sim,i}\right)}^{2}\sum_{i=1}^{n}{\left(Q_{obs,i}-{\bar{Q}}_{obs,i}\right)}^{2}}}\right]}^{2}$$

The value of R^2^ was calculated using Formula (4). Similar to NSE, the closer the value of R^2^ was to 1, the simulation results were better. The PBIAS value evaluated the relationship between simulated values and measured values, and determined whether they were overestimated or underestimated. A PBIAS value was greater than 0, which indicates a model underestimation bias; PBIAS < 0 indicates a model overestimation bias. The value of PBIAS was obtained from Formula (5).
(5)$$PBIAS=\frac{\sum_{i=1}^{n}(Q_{sim,i}-Q_{obs,i})}{\sum_{i=1}^{n}Q_{obs,i}}\times 100$$

As with NSE, PBIAS was converted into four levels to evaluate simulation performance. The simulation performance was classified as either “unsatisfactory” (±25% ≤ PBIAS), “satisfactory” (±15% ≤ PBIAS < ±25%), “good” (±10% ≤ PBIAS < ±15%), “very good” (PBIAS < ±10%), respectively. $Q_{\mathrm{obs},i}$ was the ith day observation (m^3^/s); $Q_{\mathrm{sim},i}$ was the ith day model simulated discharge (m^3^/s); ${\bar{Q}}_{\mathrm{sim},i}$ and ${\bar{Q}}_{\mathrm{obs},i}$ were the average simulation and observations (m^3^/s), respectively; and n was the total number of the observations. 

Sensitivity analysis was employed to determine how the model parameters influenced the simulation results [82]. The sequential uncertainty fitting (SUFI-2) algorithm [82] was applied, which considered many uncertainties, including the input data, model structure, parameters, and observations, in order to obtain the sensitivity results in this study. The global sensitivity was the main method used to conduct the sensitivity analysis. The *T*-states and *p*-values of the parameters were used before and after improving the statistical model to obtain parameter sensitivity [78,79,80,83]. The global sensitivity method was applied for sensitivity analysis. During the calibration process, this method was used to test the sample with a *T*-state hypothesis; compared with the critical value, the larger the better. The *p*-value was the *p* probability value corresponding to the *t*-test value in Table 1; the *p*-value reflected the significance of the *T*-state. In this method, the *T*-state was used as the sensitivity reference. The greater the absolute value of the *T*-state, the higher the sensitivity. At the same time, the *p*-value was used to indicate the significance of the *T*-state. The closer the *p*-value of the parameter was to 0, the greater its significance.

## 4. Results

### 4.1. Effects of Parameters on the Modified Daily and Sub-Daily Models

To analyze the changes and effects of parameters on the simulations, the original and newly added model parameters were calibrated, as shown in Table 1. From the model parameter library, an additional 25 sensitive parameters were selected to participate in the calibration of the model and two new parameters were added: Snowfall accumulated temperature (SFTMP_accu) and snowmelt accumulated temperature (SMTMP_accu). On the basis of the snowfall and melting temperatures in the original model, the accumulated temperature method was applied for the accumulated temperature range. From the calculations, the range of the snowfall accumulated temperature and snowmelt accumulated temperature were determined to be 0–40 °C. When calibrating the parameters of the model on both the daily and sub-daily scales, some parameters exhibited significant changes. For example, the SCS runoff curve number (CN2), the threshold depth of water in the shallow aquifer required for return flow to occur (GWQMN), the groundwater delay (GW_DELAY) and the initial depth of water in the shallow aquifer (SHALLST), precipitation lapse rate (PLAPS), and effective hydraulic conductivity in tributary channel alluvium (CH_K1) varied by more than three during the calibration process, which may have been due to the difference in the hydrological simulation process between the sub-daily model and the daily model. The SHALLST parameter changed as much as 91 mm. The remaining parameters displayed little change. Some parameters remain unchanged, such as Manning’s “*n*” value for the main channel (CH_N2). The new parameter SFTMP_accu changed from 24 on the daily scale to 26 on the hourly scale, while the corresponding SMTMP_accu change was from 18 to 19. In general, the parameters with large variations were considered influence the sub-daily model simulation significantly.

Thirty parameters were selected to participate in the calibration of the model. In order to obtain the sensitivity information of each parameter in the model, the model was simulated 1000 times to obtain the sensitivity information of the parameters. The global sensitivity analysis method is adopted in this study. There are detailed results regarding T-states and *p*-values for global sensitivity shown in Table 2. There were many parameters show high sensitivity, the T-states of effective hydraulic conductivity in main channel alluvium (CH_K2) was 51.75 and the *p*-values was 0.00, that means CH_K2 was the most sensitive parameter. In addition, the parameters related to snow melting, such as PLAPS, SMTMP, maximum melt rate for snow during the year (SMFMX), minimum melt rate for snow during the year (SMFMN), SFTMP, temperature lapse rate (TLAPS) also had high sensitivity. The new added parameters, SMTMP_accu, SFTMP_accu T-states values were 26.23 and −4.96 respectively, and they also had high sensitivity.

### 4.2. Daily Simulation Results

In order to demonstrate the effect of the simulation results before and after the model modification, the simulation results of the calibration (2013) and validation (2014) period models were compared with the measured data. To highlight the performance of different flood types before and after the model modification, the precipitation data and runoff data were analyzed using superposition. As shown in Figure 5, the precipitation in the TRB was primarily concentrated from April–September. For the March calibration, there was less precipitation, but the runoff showed a sudden increase. The original model simulated this event too early. At the same time, there was a large deviation in the size of the runoff. The modified model, however, simulated the event in both time and flow more accurate. In the summer, which was dominated by rainfall-type floods and mixed floods [11], the original model was usually inaccurate in its simulation of flood peaks. Although the modified model shifted some of the simulated flood processes earlier, its performance of flood peaks was better than the original model. During the verification period, the advantage of the modified model for the flood process simulation was more evident. Although the original model guaranteed the value of runoff to a certain extent, its performance accuracy for the flood peaks was poor. Unlike the original model, the modified model was better able to distinguish flood processes, and its flood peak performance was more accurate. The modified model was also superior to the original model in the simulation of the base flow.

The simulation results were compared before and after model modification; the simulation accuracy is presented in Table 3. The respective NSE values increased from 0.71 and 0.64 in the calibration and validation periods to 0.75 and 0.69, and from 0.66 to 0.7 in the overall study period (2013–2014). When R^2^ was used for precision evaluation, the model’s regular performance was not as apparent, but in the validation and study periods, the R^2^ values rose from 0.75 to 0.81 and from 0.8 to 0.84, respectively. The simulation results of the original model did not perform well in the evaluation index of PBIAS. The model improved from −18.04 to 2.89 in the validation period and from 7.3 to 6.79 in the overall study period.

### 4.3. Sub-Daily Simulation Results

On the basis of the daily scale, the hourly scale simulation was also carried out. The highlight of this study was the development of high temporal resolution flood process simulation. Therefore, during the high flood season (May–September), a relatively obvious flood process for each month was selected to demonstrate the simulated effect of the modified model. The flood process simulation results during the calibration are shown in Figure 6. The duration of the flood processes generally lasted about one day. In May, the original model simulated the flood peak process too early, and the simulation of the peak also had a large error. The improved model simulated the flood peak process more accurately, with higher temporal and magnitude accuracy.

During the validation period (Figure 7), the modified model also performed well in the flood process simulation. For the flood simulation processes in May, the original model did not shift the simulation of the flood processes late enough, although the simulation of the flood peak was reasonable. In the simulation of the flood simulation processes in June, the original model significantly underestimated the flood magnitude, and the flood peak again appeared ahead of schedule. The performance of the modified model was consistent with the measured data, however, the simulation of the flood peak value was more accurate. For the flood simulation process in July, the original model significantly overestimated the flood peak and performed poorly in terms of temporal accuracy. The modified model was closer to the measured flood peak magnitude and was more temporally accurate than the original model. The original model simulation exhibited both a significant underestimation and a significant time advancement of the flood peak in August. These inaccuracies, however, did not exist in the modified model. Although the August peak was slightly overestimated, it was more accurate in comparison. For the September flood simulation processes, there was a sharp increase in discharge and an expanded flood time. For these reasons, the simulation of the original model was unsatisfactory, both in terms of the simulation of flood process and flood peak. Performance of the modified model, however, was more accurate.

In order to compare the model’s performance of flood peak value and time of occurrence before and after model modification, the flood peak values and time errors of the model simulation were calculated and compiled. The flood peak error was obtained by subtracting the simulated value from the measured value, while the time error primarily involved comparing the simulated and measured times at which the flood peaks appeared. During the calibration period, the model simulation underestimated the flood peak. In the flood event of 19 May, the simulation error of the original model was 12.52 m^3^/s, and the time error was 11 h ahead. After model modification, the peak error was reduced to 2.4 m^3^/s, and the time error was improved to one hour. On 25 June, the flood peak error of the original model was 69.39 m^3^/s, whereas that of the improved model was only 4.85 m^3^/s. In terms of the time error, the simulation error of the original model was five hours ahead, while that of the modified model was 0 h, i.e., it was exactly right. On 4 July, the time of the flood peak of the original model was more accurate, but the error of the peak value of the simulation was larger. Following model improvement, the flood peak error decreased from 87.58 m^3^/s to 8.54 m^3^/s. On 4 August, the original model’s flood peak error was as high as 107.29 m^3^/s, but the improved model’s error was reduced to 32.77 m^3^/s, and the time error was reduced from four hours to one hour. For the flood processes on 7 September, the original model error was 23.41 m^3^/s, but the simulation error of the improved model was overestimated by only 0.92 m^3^/s, and the time of the flood peak was consistent with the observed value.

During the verification period, the flood peak simulation of the model was more overestimated than that of the calibration period. On 21 May, the flood peak error of the original model was −5.9 m^3^/s while that of the modified model was −7.65 m^3^/s. Although the accuracy of the original model was higher than that of the modified model, the modified model was consistent with the measured data in terms of time. On 10 June, the flood peak error of the original model was 49.14 m^3^/s, exhibiting a large underestimation. The modified model decreased the error to 4.24 m^3^/s, and the time accuracy also improved from two hours to consistent with the measured data. On 8 July, the simulation error of the original model reached its maximum, which was −451.5 m^3^/s, displaying a serious overestimation. After modification, the error decreased to only −8.56 m^3^/s. In terms of time accuracy, the original model exhibited a premature flood peak for this event, while the improved model increased the time accuracy from four hours to two hours. On 15 August, the modified model effectively improved the flood peak temporal error from 11 h to 0, and the error of the peak value decreased from 39.84 m^3^/s to 12.58 m^3^/s. On 8 September, the modified model decreased the error of the flood peak from 70.77 m^3^/s to 10.43 m^3^/s and significantly delayed the peak time.

In order to reflect the effect of model modification, all of the flood events from April–September were statistically verified and compared using the measured data. For the 2013 calibration period, there were a total of 14 flood events: Two in April, one in May, four in June, three in July, three in August, and one in September. For the 2014 validation period, there were a total of 12 flood events: Two in May, three in June, two in July, four in August, and one in September. The performance of the original model simulation is presented in Figure 8. Figure 8a shows that the flood events were significantly underestimated when compared with the measured values. The simulation results of the modified model were closer to the measured values, and the simulation accuracy was significantly improved temporally. Figure 8b compares the simulation results for the validation period. The accuracy of the modified model’s simulation results was somewhat improved, with the simulation results closer to the actual observations.

## 5. Discussion

### 5.1. Model Modification

The snowmelt module of the original model was calculated primarily with the degree day factor method. The daily average temperature was the only simple snowmelt determination condition [84]. Snowmelt events occur when the temperature exceeds a set threshold. If the average daily temperature is used to determine the type of precipitation and snowmelt conditions, the model will have significant limitations [6,85,86,87]. The precipitation in mountainous areas is influenced by both topography and microclimates [88]. If the average daily temperature is used to determine the type of precipitation, it will produce an error. Similarly, this situation exists in the calculation of snowmelt. In mountainous areas, accumulated temperature is an indispensable parameter for the calculation of precipitation patterns and snowmelt [89,90]. Therefore, it is necessary to improve the traditional day-factor method in order to enhance the performance of the accumulated temperature’s influence on precipitation type and snowmelt.

Following model modification, the average accuracy of the determination of precipitation patterns was as high as 86.88%. This demonstrates that the introduction of accumulated temperature restriction conditions effectively improved the accuracy of the model input data. Simulations on a daily scale revealed that the contribution of snowmelt to runoff calculated by the modified model increased by approximately 10% compared with the original model. This is mainly because the original model only uses the daily average temperature as the criterion of snow melting, but when the temperature changes greatly in a day, the phenomenon of snow melting will also be caused by long-term temperature accumulation, but the daily average temperature at this time may not reach the temperature of snow melting conditions, the traditional method cannot recognize the situation of snow melting.

The sub-daily simulation was based on the daily scale. Therefore, better daily simulation results were guaranteed through model modifications. For the runoff simulation in mountainous areas, the accurate calculation of snowmelt could improve the simulation accuracy of flood processes, especially during the spring season with high incidence of snowmelt floods. The original model did not perform well in the calculation of snowmelt, but after modifications, the calculation of the amount of snowmelt improved greatly (22.84%). In spring, the daily temperature changes rapidly and the temperature is lower. At noon, the temperature rises rapidly, which has a direct effect on snow melting and is easy to cause floods. However, the daily average temperature is the average value of a day. Through the average, the actual situation of higher temperature at noon is neglected, which has an impact on the calculation of snow melting, which can be effectively improved by model modification.

### 5.2. Model Performance

Flood types can generally be divided into rainfall, snowmelt, and mixed floods [4,6,7,91]. Mixed floods are usually accompanied by snowmelt and precipitation, both of which contribute to the flood events [88]. Therefore, it can be said that flood events are generally triggered by both rainfall and snowmelt [66]. In the detailed study of flood processes, an appropriate time scale should be selected. When selecting the scale, many factors, such as precipitation, geographic characteristics, and topography, should be taken into account. The TRB is located in the Kunlun Mountains and receives less precipitation, especially in spring. Therefore, when studying the flood processes, the main time scale was one hour. In the hourly scale simulation, the model was first calibrated on the daily scale, and thus required the simulation accuracy of the daily scale model.

The simulation results were analyzed on a daily scale using precipitation data, as shown in Figure 5. The precipitation increased significantly beginning in April, and the corresponding runoff also increased significantly in calibration time. In March, runoff had also increased, although there was no significant precipitation at that time. Combined with the topography of the basin, it was determined that the runoff supply in March originated primarily from snowmelt. By modifying the day-degree factor and increasing the accumulated temperature condition, simulation runoff suddenly increased, becoming closer to the measured data, and indicating the existence of a snowmelt flood. The modified model was closer to the observations both in terms of the simulation of base flow as well as peak flow. In summer, when there was more precipitation, more flood events were affected by the precipitation events. The original model was not ideal for the simulation of flood processes and peak flow, while the modified model was more accurate. After September, both precipitation and runoff decreased, mainly because mountain temperatures were lower and snowmelt gradually disappeared; therefore, the runoff recharge gradually diminished.

The modified model performed better during the validation period and was similar to that of the calibration period. During this period, the original model did not accurately simulate flood processes, especially the number and size of flood peaks. When heavy precipitation events occurred, the flood peak process was not apparent in the original model’s simulation. The modified model, however, performed better in terms of the timing and size of flood peaks. Notably, in May, when snowmelt was the main runoff recharge, the simulation of snowmelt flooding was more accurate. After September, since the runoff recharge source gradually decreased and disappeared, the runoff simulation exhibited a decreasing trend. In this respect, the modified model was more accurate. This is mainly due to the addition of the accumulated temperature and maximum temperature constraints, which avoided the unrealistic snowmelt amount calculated using average temperature in the original model. This reduced the unrealistic snowmelt replenishment to the runoff, thus yielding results that were closer to the observations.

During both the validation and calibration periods, the modified model provided a better simulation of runoff and flood processes than the original model. Although the overall simulation was better, some details were deficient. For example, certain lag phenomena existed in the simulation of the individual flood processes, which may have been related to such factors as data accuracy and soil interception [47,92]. 

The TRB has special climatic and geographic features, such as little vegetation and numerous rocks. When the sun rises, the surface temperature increases rapidly. When the sun sets, the temperature drops quickly. Therefore, the temperature difference between day and night is relatively large, which also causes a special snowmelt phenomenon in the mountainous area—i.e., the “one day, one peak” phenomenon [64,66,93]. As the temperature increases during the day, the snow slowly begins to melt, and the recharge to the river gradually increases, resulting in snowmelt flooding. At night, the temperature drops sharply, the melting snow gradually decreases or even stops, and freezing sometimes occurs. This process usually lasts for about 24–28 h. Therefore, the best time scale for studying the flood processes in the Kunlun Mountains is one hour. Using the daily scale as the starting point, few hourly scale studies have been conducted. As a result, demand is high for these data and models. Therefore, this sub-daily (i.e., hourly) flood processes study was carried out on the basis of the daily scale. During the flood processes from May–September, the most obvious event for each month was selected for analysis.

For simulations on a sub-daily scale, the simulation results before and after modification are presented in Figure 6 and Figure 7, respectively. Starting in early May, snowmelt increased and gradually entered a period of high flood incidence. The more obvious and larger flood events from each month were selected for simulation comparison. The flood processes usually lasted for about one day. The river runoff gradually increased beginning at around 8 a.m., mainly because as the sun rose, the temperature also gradually rose, and the snow began to melt into the river. River runoff continued to increase until about 4 p.m., when the runoff reached its maximum value and the flood peak occurred. The flood peak flow reached its maximum value at this time. After that, as the temperature and accumulated temperature gradually decreased, the amount of snowmelt and the runoff gradually decreased. This short-duration phenomenon is generally considered to be a complete flood event, and is more affected by temperature. In 2013, the comparison of flood peak values for five months revealed that the values in May and September were relatively small, whereas those that occurred from June–August were relatively large. The relatively low temperatures in May and September had a certain impact on snowmelt. During the summer months from June–August, the temperatures in the mountainous areas are relatively high during the day, which may increase the amount of snowmelt. The maximum peak flow of 283 m^3^/s occurred in August. Combined with the influence analysis of accumulated temperature, the monthly average value in August was 53.77 °C, which was the maximum for the months in the study period. During the validation period, the maximum peak flow of 272 m^3^/s occurred in July, whereas the accumulated temperature in July was 54.25 ℃, which was also the maximum value. The size of the flood peak was greatly affected by accumulated temperature under certain conditions through its influence on the amount of snowmelt.

Table 4 and Table 5 compare the model’s performance before and after modification on the sub-daily scale, respectively. The necessity of model modification was analyzed using the peak flow error and time error of each flood peak. During calibration, the flood peak error on the hourly scale was greatly reduced by the modification. The biggest improvement in the accuracy was achieved in May, when the flood peak accuracy improved by 96.07% and the time error was reduced from 11 h to one hour. This period had a high occurrence of snowmelt flooding, and fully illustrates the importance of the model improvement for flood process simulations at high time resolution. During the validation period, the biggest improvement in flood peak accuracy was achieved in July, with an increase of 98.1%. The biggest improvement in time accuracy came in August, when it increased by 11 h. Both the flood peak value and time accuracy were greatly improved by the model modification, which provided a new way to simulate flood processes and established a foundation for flood forecasting.

So as to clearly illustrate the modification results of the model, the peak flow values of flood events simulated by both models were contrasted to the observations, as shown in Figure 8. By comparing the spatial relationship between the simulated value of the model and the measured line, the simulation effect of the model can be judged. It was found that the effect of the modified model was overt, and the simulated values of the improved model were closer to the measured values. During the validation period, although the effect was not as obvious as that seen during calibration, a certain improvement was found. This may be related to the calibration of parameters and the accuracy of data.

### 5.3. Sensitivity and Uncertainty Analysis

In the original model, the top 10 sensitive parameters were CH_K2, theLateral flow travel time (LAT_TIME), SMFMX, minimum snow water content that corresponds to 100% snow cover (SNOCOVMX), TLAPS, CH_K1, Snow pack temperature lag factor (TIMP), Groundwater “revamp” coefficient (GW_REVAP), Manning’s “*n*” value for overland flow (OV_N), and base flow alpha factor (days) (ALPHA_BF). In the modified model, the top 10 sensitive parameters were LAT_TIME, PLAPS, CH_K2, ALPHA_BF, SFTMP, TLAPS, SMTMP_accu, SMFMX, soil evaporation compensation factor (ESCO), and SFTMP_accu. Thus, the sensitivity of the modified model was different from that of the original model, and the newly added parameters (SMTMP_accu, SFTMP_accu) in the modified model also exhibited more sensitivity. During the sub-daily model simulation based on the original daily model, the top 10 sensitive parameters were PLAPS, LAT_TIME, available water capacity of the soil layer (SOL_AWC), CH_K2, SNOCOVMX, saturated hydraulic conductivity (SOL_K), RCHRG-DP, SMFMX, TLAPS, and CH_K1. In addition, for the sub-daily model based on the modified daily model simulation, the top 10 sensitive parameters were PLAPS, TLAPS, CH_K1, SFTMP, CN2, SMTMP_accu, SMFMX, SFTMP_accu, SMTMP, and TIMP. After analyzing the parameter sensitivity before and after model modification, it was found that the sensitivities of parameters related to snowmelt were greatly improved after model modification, including those of PLAPS, TLAPS, SFTMP, SMTMP_accu, SMFMX, SFTMP_accu, and SMTMP. The TRB experiences significant snowfall and contains many high-altitude mountainous areas. Therefore, snowmelt provides greater recharge to the river channel, reflecting the necessity for and rationality of the model improvement. When the model calibration was corrected, the optimal parameter combination was more consistent with the actual situation of the basin and was also more convincing, as confirmed by the model simulation results.

When the SWAT model has been used to conduct relevant research on hydrological processes, the uncertainties of the simulation results have been divided into three types: The uncertainty of model structure, the uncertainty of parameters, and the uncertainty of input data [94]. Some studies [95,96,97,98,99,100,101] have shown that the uncertainties of the simulation results mainly stem from the model parameters uncertainty and the input data during the flood period. Throughout the study period, the uncertainty of the model structure and the input data could not be ignored.

## 6. Conclusions

On the basis of the original SWAT model, this study modified the traditional day-factor method for calculating snowmelt and added the temperature condition on precipitation morphology and snowmelt conditions. As a result of these modifications, the simulation accuracy of the model on a daily scale significantly improved, especially in the spring, when snowmelt runoff was the main source of replenishment. As a result, the simulation of the snowmelt flooding process in spring was more obvious and accurate than that of the original model, thus compensating for the original model’s shortcomings in the simulation of snowmelt flooding. 

As a result of these improvements, the NSE value increased from 0.71 to 0.75 of the calibration period, and in the validation period it increased from 0.64 to 0.69. During the overall study period, the NSE value increased from 0.66 to 0.7, the R^2^ value increased from 0.8 to 0.84, and the PBIAS changed from 7.3 to 6.79, the phenomenon of model underestimation has been improved. In addition, when the modified model was adopted to improve the snowmelt calculation, snowmelt volume increased by 22.84% and the contribution to the channel volume increased by 10%. During the spring, which had less precipitation and for which snowmelt was the main channel recharge, the model’s simulation accuracy was improved and the simulation of snowmelt flooding was more prominent in the overall simulation of the flood event processes.

The highlight of this study was the sub-daily scale flood process simulation performed on the basis of the daily scale, along with the analysis of the model’s performance on the hourly scale before and after modification. From the simulation and comparison of flood events, it was found that both the magnitudes and times of the flood peaks appearing at the hourly scale of the modified model have been greatly improved, also proving that the flood process simulation at a high time resolution was more accurate following model modification. The results also revealed that it is feasible to simulate flood processes on a daily or sub-daily scale with high temporal resolution by modifying the day-degree factor technique, thus providing a new reference method for flood process simulations. This study encourages new ideas for the investigation of flood process changes at a high time resolution and provides a new flood-forecasting method. This work is of great significance in the attempt to save lives and ensure the safety of property through flood relief and natural disaster prevention.

## Figures and Tables

**Figure 1 ijerph-16-03118-f001:**
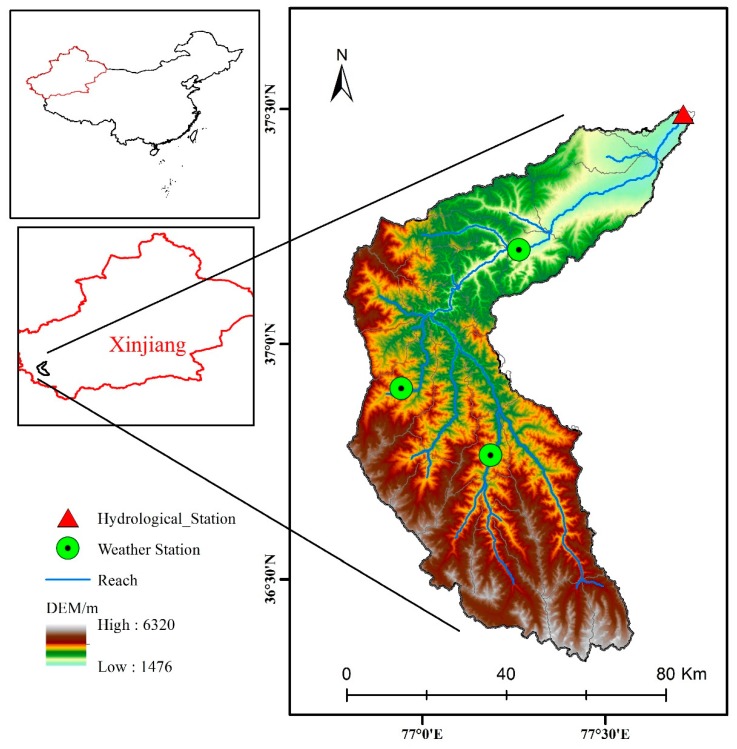
Topography and sites distribution of the Tizinafu River Basin (TRB), Kunlun Mountains, West China.

**Figure 2 ijerph-16-03118-f002:**
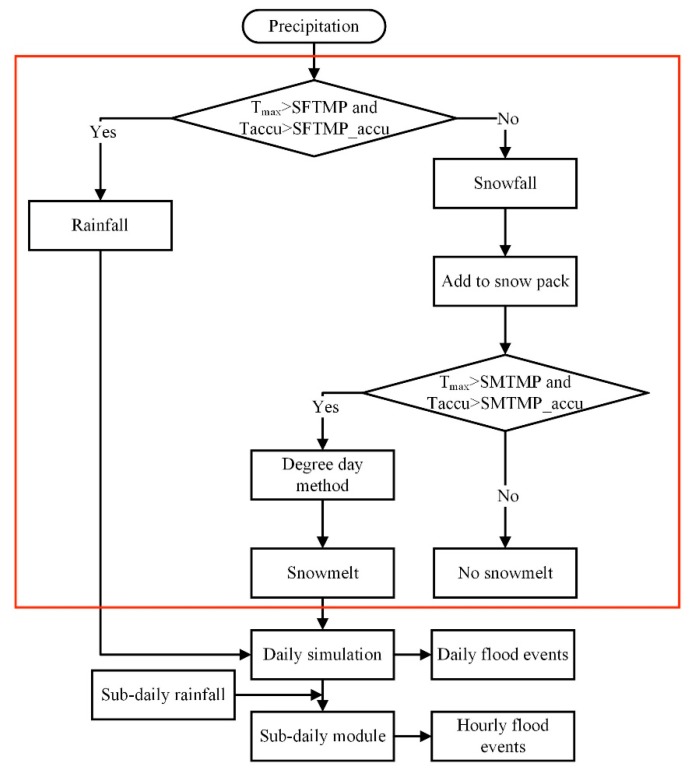
Flowchart of flood process modeling ideas.

**Figure 3 ijerph-16-03118-f003:**
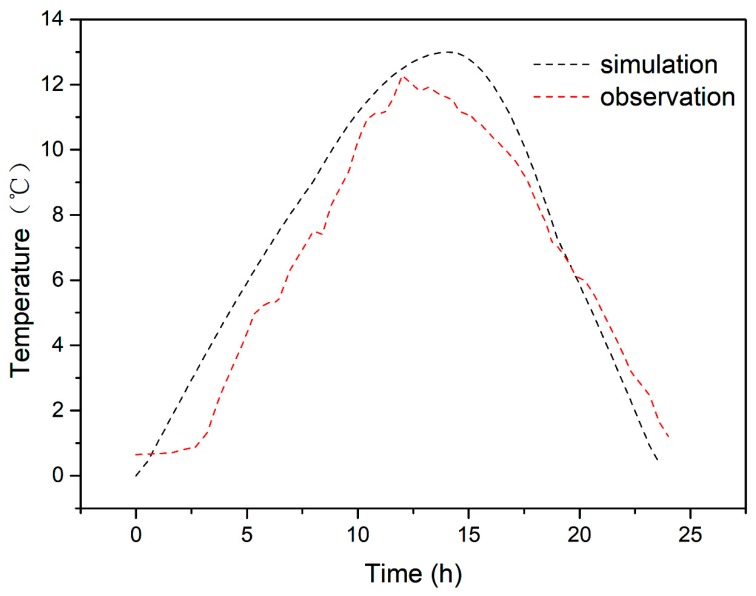
Daily actual temperature change and simulation curve.

**Figure 4 ijerph-16-03118-f004:**
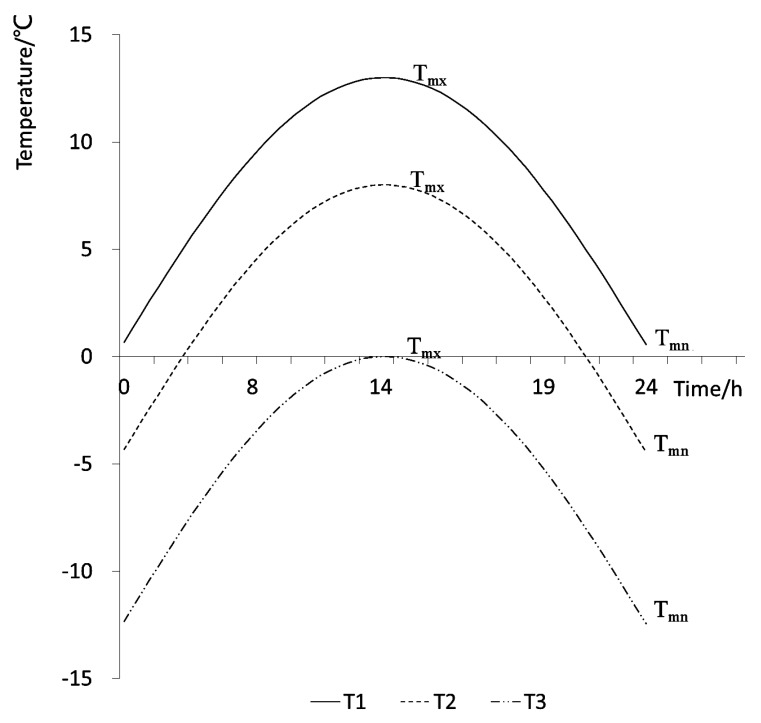
Three different daily temperature variations.

**Figure 5 ijerph-16-03118-f005:**
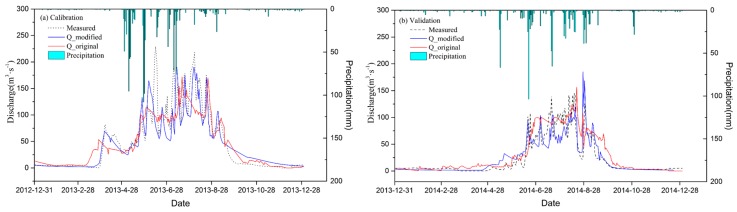
Simulation results of the model on the daily scale: (**a**) Represents the comparison of precipitation, simulation, and observation during the calibration period; (**b**) represents the comparison of precipitation, simulation, and observation during the validation period.

**Figure 6 ijerph-16-03118-f006:**
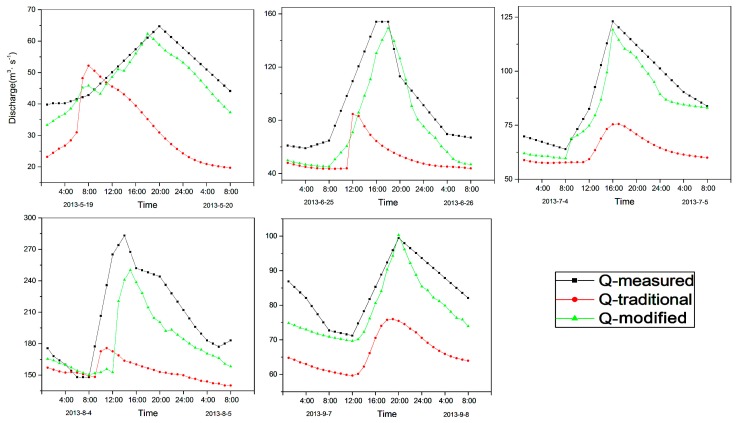
Simulation results of the improved model on an hourly scale during the calibration period.

**Figure 7 ijerph-16-03118-f007:**
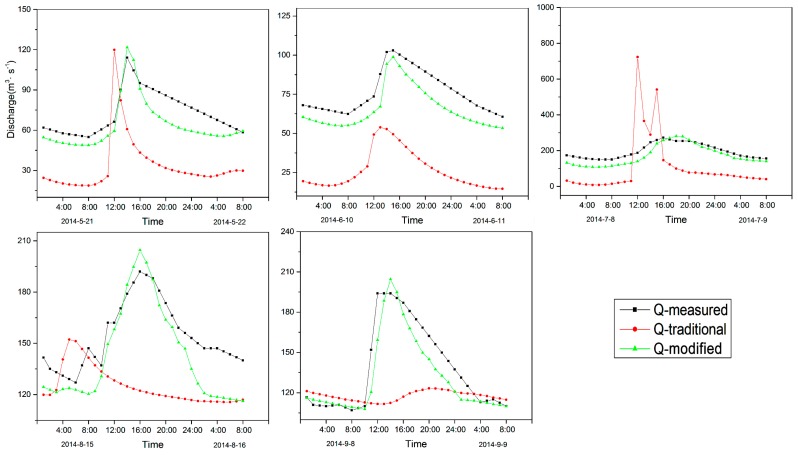
Simulation results of the improved model on an hourly scale during the validation period.

**Figure 8 ijerph-16-03118-f008:**
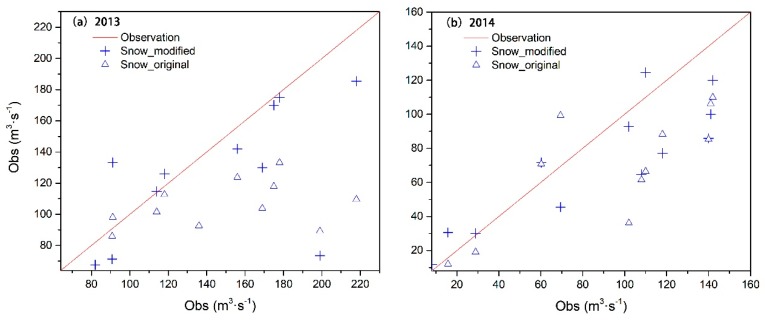
All of the flood peaks simulated by the model compared with the observations: (**a**) Comparison between simulated flood peak values and observations before and after model modification during the calibration period; (**b**) comparison between simulated flood peak values and observations before and after model modification during the validation period.

**Table 1 ijerph-16-03118-t001:** Important parameters in the model calibration process.

Parameter	Description	Lower Bound	Upper Bound	Daily Simulation Calibrated Value	Sub-Daily Simulation Calibrated Value
CN2	SCS runoff curve number	35	98	72.8	68.98
ALPHA_BF	Base flow alpha factor (days)	0	1	0.16	0.15
GW_DELAY	Groundwater delay (days)	0	500	216.6	223.72
GWQMN	Threshold depth of water in the shallow aquifer required for return flow to occur (mm)	0	5000	742.7	713.68
SHALLST	Initial depth of water in the shallow aquifer (mm)	0	50000	4926	4835
GW_REVAP	Groundwater “revamp” coefficient	0.02	0.2	0.05	0.04
SOL_K	Saturated hydraulic conductivity	0	2000	830.1	826.54
SOL_AWC	Available water capacity of the soil layer	0	1	0.33	0.28
SFTMP	Snowfall temperature	−20	20	3.24	3.05
SMTMP	Snowmelt base temperature	−20	20	2.97	2.76
SMFMX	Maximum melt rate for snow during the year	0	20	7.87	7.65
SMFMN	Minimum melt rate for snow during the year	0	20	9.49	8.86
TIMP	Snow pack temperature lag factor	0	1	0.54	0.53
SNOCOVMX	Minimum snow water content that corresponds to 100% snow cover	0	500	66.1	65.1
SURLAG	Surface runoff lag time	0.05	24	11.17	11.02
PLAPS	Precipitation lapse rate	−20	20	21	19
TLAPS	Temperature lapse rate	−10	10	−7.31	−7.8
CH_N1	Manning’s *“n”* value for the tributary channels	0.01	30	5.77	5.87
CH_K1	Effective hydraulic conductivity in tributary channel alluvium	0	300	299.45	279.34
OV_N	Manning’s *“n”* value for overland flow	0.01	30	11.63	11.54
ESCO	Soil evaporation compensation factor	0	1	0.37	0.36
EPCO	Plant uptake compensation factor	0	1	0.37	0.35
CH_N2	Manning’s *“n”* value for the main channel	−0.01	0.3	0.02	0.02
CH_K2	Effective hydraulic conductivity in main channel alluvium	−0.01	500	49.53	48.75
SNO_SUB	Initial snow water content	0	150	95.43	98.37
SFTMP_accu	Snowfall accumulated temperature	0	40	24	26
SMTMP_accu	Snowmelt base accumulated temperature	0	40	18	19

**Table 2 ijerph-16-03118-t002:** The parameter information for global sensitivity.

Parameter Name	T-States	*p*-Value
CH_K2	51.75	0.00
PLAPS	21.82	0.00
LAT_TTIME	29.93	0.00
SMTMP_accu	26.23	0.00
SMTMP	22.23	0.00
SMFMX	10.90	0.01
SMFMN	8.24	0.03
SOL_K	5.20	0.03
SOL_AWC	2.09	0.04
ESCO	1.52	0.13
SURLAG	1.41	0.16
TIMP	1.30	0.20
SNO_SUB	1.01	0.31
EPCO	0.54	0.59
REVAPMN	0.43	0.67
GWQMN	0.25	0.80
SMFMN	0.24	0.81
RCHRG_DP(Deep aquifer percolation fraction)	0.17	0.86
CH_N2	0.05	0.96
CH_N1	0.00	1.00
SNOCOVMX	−0.08	0.94
SHALLST	−0.76	0.45
GW_REVAP	−1.33	0.18
GW_DELAY	−1.76	0.08
CN2	−1.79	0.08
OV_N	−2.70	0.05
CH_K1	−3.19	0.05
SFTMP	−4.01	0.05
SFTMP_accu	−4.96	0.04
TLAPS	−5.31	0.02
ALPHA_BF	−6.13	0.00

**Table 3 ijerph-16-03118-t003:** The results of daily calibration and validation for TRB, Kunlun Mountains, West China.

Period	NSE		R^2^		PBIAS (%)	
	Original	Modified	Original	Modified	Original	Modified
Calibration (2013)	0.71	0.75	0.89	0.89	5.79	7.3
Validation (2014)	0.64	0.69	0.75	0.81	−18.04	2.89
Overall (2013–2014)	0.66	0.7	0.8	0.84	7.3	6.79

**Table 4 ijerph-16-03118-t004:** Simulation results of flood peak before and after model modification during the calibration period.

Date	Original Flood Peak Error (m^3^·s^−1^)	Modified Flood Peak Error (m^3^·s^−1^)	Original Flood Peak Time Error (h)	Modified Flood Peak Time Error (h)
19 May 2013	12.52	2.4	11	1
25 June 2013	69.39	4.85	5	0
4 July 2013	87.58	8.54	−1	−1
4 August 2013	107.29	32.77	4	−1
7 September 2013	23.41	−0.92	1	0

**Table 5 ijerph-16-03118-t005:** Simulation results of flood peak before and after model modification during the validation period.

Date	Original Flood Peak Error (m^3^·s^−1^)	Modified Flood Peak Error (m^3^·s^−1^)	Original Flood Peak Time Error (h)	Modified Flood Peak Time Error (h)
21 May 2014	−5.9	−7.65	2	0
10 June 2014	49.14	4.24	2	0
8 July 2014	−451.5	−8.56	4	−2
15 August 2014	39.84	−12.58	11	0
8 September 2014	70.77	10.43	−8	−2

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
