# Peer review of "Sub-Daily Simulation of Mountain Flood Processes Based on the Modified Soil Water Assessment Tool (SWAT) Model"

_ijerph, 2019, doi:10.3390/ijerph16173118_

Round 1
Reviewer 1 Report
Given that the snowmelt module in SWAT model has not considered the influence of accumulated temperature, the author modified this module by taking the accumulated temperature into account and study its performance in high-altitude mountainous areas regarding daily and hourly flood event simulation. In general, this manuscript is well-structured. However, some improvements and clarifications are needed before the manuscript published. Although the revision is minor, I expect the authors to address all of my comments and suggestions that are listed below:
Detailed comments:
1. 224 line: In Figure 4, there are Tmax and Tmin. But in equation (1), the authors used Tmx, and Tmn which is supposed to be the same as shown in the Figure 4. My suggestion is either change the symbols in Figure or change them in all the relevant equations.
2. Line 229: t is the radian corresponding to the moment of the day, and the and the radians corresponding to 0. How to obtain these radians which are corresponding to 0? Because the method of SFTMPaccu calculation is the main part in this study, and these two radians are one of key factors to determine SFTMPaccu. It is better to explain a bit more about how to obtain these two radians in methodology part.
3. Line 248 – 250: Based on equation (5), if PBIAS < 0 , it means Qsim < Qobs which indicates the simulation is underestimated. However, this conclusion is the opposite to the conclusion in line 248.
4. Line 253-254: What is the scale of the average simulation and average observation? If they are the daily averages, what is the difference between and ?
5. Line 516: The sequential uncertainly fitting (SUFI-2) algorithm was applied for sensitivity and uncertainty analysis. It may be better if the authors explain a bit more about this algorithm in methodology part. Because this method only appears once in the manuscript, which is in discussion part. This is not that logical for me.
6. Line 518 – 520: There is no detailed results regarding T-states and p-values for global sensitivity. I can only find general conclusion regarding this part in discussion section. Because sensitivity analysis is one of the parts in methodology, it is supposed to have its relevant detailed results in result section instead of only conclusion in discussion section.
General comments:
7. SMTMP_accu calculation:
In SWAT, there is a parameter called Tsnow, describing the temperature of snow which is used to calculate the snowmelt. The calculation for this parameter depends on the mean temperature on a given day and snowpack temperature on the previous day. In my opinion,
Hourly accumulated temperature (T_accu) can influence snowpack temperature as well and has the impact on the snowmelt. Therefore, I suggest authors also concern snowpack temperature at the previous hour and include it in the methodology part.
8. Because in SWAT, the snowmelt module is on a daily basis. When a flood event was simulated by adding hourly rainfall driven data, how about the snowmelt process? Is it also on an hourly scale? If yes, how to deal with other parameters on a daily basis for snowmelt calculation, i.e. SNOCOV, bmlt, ℓsno. If not, how to guarantee the accuracy of the hourly flood event simulation in mountainous area? In my opinion, in these areas, most of the floods are triggered by sudden snowmelt instead of rainfall. Therefore, I suggest the authors may not only include hourly rainfall driven data but also consider the snowmelt other relevant parameters on an hour scale in the sub-daily module to simulate the hourly flood events.
In summary, I suggest author to revise this manuscript based on comments and resubmit as new.

Author Response
Dear reviewer and editor:
Thank you very much for your valuable comments and kind suggestions on our submission.Please see the attachment.

Reviewer 2 Report
I think that the manuscript is well written
I have only one comments, concerning references about uncertainty evaluation (i.e. 91-94 citations into the manuscript): I suggest to increase the citations with the following papers:
https://doi.org/10.1016/j.jhydrol.2017.06.004
https://doi.org/10.1016/j.jhydrol.2012.11.019
https://doi.org/10.1016/j.jhydrol.2014.07.049 https://doi.org/10.1016/j.jhydrol.2013.10.055Author Response
Dear reviewer and editor:
Thank you very much for your valuable comments and kind suggestions on our submission.Please see the attachment.

Reviewer 3 Report
I suggest that the authors need to explain the sub-daily simulation more understandable in methodology section. Also, the results found in this study need to be discussed more physically what happened in hourly simulation compared daily simulation. Please check the follow comments.
comments:
n Page 3, Lines 104~121: This paragraph describes site information in China. That can be moved to Study area section or mentioned early in Introduction section after summarized. Also, I suggest to add more related researches for sub-daily simulation how it works in SWAT.
n Introduction: Please make the objectives of this study clear.
n Study area: I suggest to state the seasonal temperature or max/min temperature in this study area.
n Page 5, Line 157: Replace “as” à “is”
n Figure 3. There is a typo in y-axis.
n Page 6, Lines 185~189: Please make the method of the hourly scale simulation clearer.
n Page 10: Except CN, all parameters were not significantly changed compared to the ranges of the parameters. Is there any physical meaning for the parameter changes between daily and hourly simulation??
n Figure 5: From the calibration graphs, the discharge on March was increase with no significant precipitation. Is that correct??
n Figure 8: Please check the names of axes.
n Page 15, Lines 400-412: This paragraph seems to repeat the introduction. Furthermore, in discussion section, authors need to discuss more physically based on the results what they found in this study, not just comparing model performances and summarizing the results.
Author Response

(The authors gave the same response as above.)
